# Assessing Urban Yellow Fever Transmission Risk: *Aedes aegypti* Vector Competence in Argentina

**DOI:** 10.3390/v17050718

**Published:** 2025-05-16

**Authors:** Estefanía R. Boaglio, Evangelina Muttis, Mariel Feroci, Cintia Fabbri, Graciela Minardi, Juliana Sánchez, María V. Micieli, Silvina Goenaga

**Affiliations:** 1Instituto Nacional de Enfermedades Virales Humanas (INEVH-ANLIS) “Dr. Julio I. Maiztegui”, Pergamino 2700, Argentina; erboaglio@comunidad.unnoba.edu.ar (E.R.B.);; 2Centro de Bioinvestigaciones (CeBio) y Centro de Investigación y Transferencia del Noroeste de Buenos Aires (CITNOBA-CONICET), Universidad Nacional del Noroeste de la Provincia de Buenos Aires, Pergamino 2700, Argentina; 3Centro de Estudios Parasitológicos y de Vectores (CEPAVE-CCT La Plata-CONICET), Universidad Nacional de La Plata (UNLP), La Plata 1900, Argentina; emuttis@cepave.edu.ar (E.M.); victoria@cepave.edu.ar (M.V.M.)

**Keywords:** yellow fever virus, Argentina, *Aedes aegypti*, vector competence

## Abstract

Yellow fever is a viral disease with historical importance since epidemics caused thousands of deaths at the end of the 19th century in Argentina. That event was associated with the presence of *Aedes aegypti*. After the mosquito eradication in South America in the 1960–1970 decade, no epidemic was detected related to this species but epizootics have occurred due to sylvatic vectors belonging to *Haemagogus* and *Sabethes* genera. Due to the recolonization of *Ae. aegypti* and its expanded distribution, the risk of the urbanization of yellow fever has increased over time. However, the reasons why the urban cycle of the yellow fever virus (YFV) has not occurred in South America so far are unknown. We explore the vector competence of *Ae. aegypti* for YFV transmission. The mosquitos evaluated belonged to colonies from center and northwest cities from Argentina, taking into account the particular genetic features of this mosquito species detected in this country from 2016. We used a viral strain originally isolated in 2009 from *Sabethes albiprivus* in the country. Viral infection in mosquito body, legs, and saliva was evaluated to estimate the rates of infection, dissemination, and transmission. Our results indicate that both mosquito colonies are competent vectors in the transmission of the YFV but with differences between them. Regarding the infection timeline, we observed a very early infection in the La Plata colony at 3 DPI in contrast to previous studies. This research improves our understanding of the risks of urban YFV transmission in Argentina, highlighting the need for surveillance and specialized vector control strategies in urban settings to prevent yellow fever outbreaks.

## 1. Introduction

The yellow fever virus (YFV) is an arbovirus that causes viral hemorrhagic fever in humans, with a notable history of morbidity and mortality, particularly in South America and Africa where, despite the availability of effective vaccines, it remains a major public health problem, with an annual incidence of around 109,000 and 51,000 deaths [1]. The etiological agent is Yellow fever virus, a prototypical member of the genus *Flavivirus* (*Flaviviridae* family). It is a positive-sense single-stranded RNA virus with a genome of approximately 11 kb [2].

Phylogenetic studies indicate that the YFV originated in Africa 1500 years ago. Bryant et al. (2007) demonstrate that the strain was introduced to the Americas about 300–400 years ago, most likely through the slave trade. Subsequently, the YFV spread westward across the continent and adapted to indigenous mosquitoes and hosts leading to a spill back into the enzootic cycle in South America [3].

In the Americas, two transmission cycles of the YFV have been recognized: the urban cycle, involving *Ae. aegypti* mosquitoes and humans, and the sylvatic cycle, involving non-human primates and various mosquito species living in forests, acting as vectors and presumably as reservoirs, particularly belonging to *Hemagogus* and *Sabethes* genera [4,5].

Urban outbreaks were relevant at the end of the 19th century, when they caused thousands of deaths in Argentina [6]. Subsequently, *Ae. aegypti* eradication in the 1960s from Argentina and other countries in the Americas has led to the elimination of the YFV urban cycle [7]. However, because the vector surveillance and control measures were not maintained for different reasons, such as economics, politics, and the discovery of lethal effects of DDT, the reinfestation of *Ae. aegypti* in South America was not long in coming [8]. However, the reason why urban outbreaks of yellow fever are not currently occurring is unclear.

The YF sylvatic cycle is endemic in the neighboring country Brazil where, during the last decades, it has left the limits of the Amazon and began to reach the center and south of the country [9], getting closer to the Argentinean provinces of Misiones and Corrientes, with repetitive outbreaks in urban and peri-urban areas from 2016 to 2021 in Brazil [10,11,12,13]. Habitat fragmentation and climate change could increase the risk of YFV transmission because they promote closeness with howler monkeys and mosquitoes, where the boundaries between the jungle and urbanization have been blurred [7].

In Argentina, the last outbreaks occurred between 2007 and 2009, confined to sylvatic environments with *Sabethes albiprivus* as the main implicated vector with significant consequences for non-human primates and five fatal human cases [14,15]. Currently, the increases in viral activity usually occur in a regional context of epizootics that affect southern Brazil and eastern Paraguay [16], but the risk of urban YF remains latent.

Several studies were conducted to evaluate vector competence for the YFV in *Ae. aegypti* using African, Asian, and American mosquito populations. A few of them were with Brazilian populations [17,18,19,20], but sometimes these studies only evaluate the infection or transmission rate around 14 days after infection. Information about the first days of infection is insufficient. Furthermore, until now, no experimental trials related to the YFV have been carried out in Argentina. In this scenario, questions remain about whether *Ae. aegypti* is still competent against the YFV and why the urban cycle does not occur in Latin American countries. In order to answer these questions, experimental assays were conducted to evaluate and compare the intrinsic abilities of *Ae. aegypti* colonies, originating from the center (La Plata city) and northwest of Argentina (Tartagal city), which infected and transmitted YFV. Our work contributes to the understanding of the transmission dynamics of the YFV in South America, more precisely in Argentina, and highlights the need for continued surveillance and vector control efforts, especially in urban areas where the resurgence of YF represents a significant threat.

## 2. Materials and Methods

### 2.1. Mosquito Strains

*Aedes aegypti* mosquitoes used in the study were obtained from eggs collected from two laboratory colonies. These colonies were established from individuals originally collected from extremes of this mosquito species’ distribution in Argentina, the northwestern subtropical region (Tartagal, Salta province) and the center temperate area (La Plata, Buenos Aires province). The La Plata colony started from larvae collected at La Plata cemetery in 2014, and the Tartagal colony was obtained from field eggs collected in Salta province during 2014. Both colonies were maintained at the Centro de Estudios Parasitológicos y de Vectores (CEPAVE) in La Plata at 25 ± 2 °C and photoperiod 12:12 (L:D).

### 2.2. Viral Strain

The viral strain used for assays was originally isolated from *Sa. albiprivus* during the last outbreak of YF in 2009 in Argentina [15].

The viral stock (7.2 log10 PFU/mL) was prepared after four passages on Vero C76 cells and frozen at −86 °C until oral mosquito infections. The titer was obtained by plaque assays on 12-well plates of Vero cells and qRT-PCR to confirm YFV infection was performed.

### 2.3. Oral Infection in Adult Aedes aegypti Mosquitoes

Oral infection was performed as previously described by Bonica et al. (2019) [21]. Briefly, adult mosquitoes were maintained under controlled conditions at 27 ± 1 °C, with a relative humidity of 70 ± 10%, and a 16:8 h light:dark cycle, and provided with raisins and water. A 24 h period of starvation preceded the oral infection. Females between five and eight days old were orally infected through a blood meal provided by an artificial glass feeder that maintains the blood temperature at 37 °C [21]. The infectious blood meals were composed of 2 mL of ovine blood (Lab Alfredo Gutierrez, C.A.B.A., Argentina), 0.2 mL (10%) of sucrose 5%, and 2 mL of previously frozen cell culture supernatant containing YFV. Titers of each infected blood meal were obtained by plaque assays for control. The mosquitoes were allowed to feed for one hour. Engorged females were counted and segregated into cardboard cages for subsequent analysis at 3, 7, 14, and 21-days post-infection (DPI) (Figure 1). The cages were supplied with raisins and water for feeding mosquitoes ad libitum and placed in an incubator, preserving the initial temperature and humidity conditions until the analysis was carried out each day post infection. All infection assays were conducted within the confines of biosafety level 3 facilities at the Instituto Nacional de Enfermedades Virales Humanas (INEVH-ANLIS) in Pergamino, Buenos Aires.

### 2.4. Mosquito Dissections and Sample Processing

At 3, 7, 14, and 21 DPI, mosquitoes were anesthetized with triethylamine as previously described [22]. Bodies, legs, and saliva from each mosquito were individualized and analyzed separately. The proboscis of each immobilized mosquito was inserted into a capillary containing 5 μL of Minimum Essential Medium (MEM) supplemented with 20% of fetal bovine serum (FBS). After 30 min of salivation, the proboscis was removed from the capillary tubes, and legs and bodies were separated into individual tubes. The contents of the capillaries were transferred to tubes containing 300 μL of MEM supplemented with 20% FBS. Each female mosquito in the study was tagged and assigned a unique identifier. All samples were stored at −86 °C until processing.

The bodies and legs were homogenized in microcentrifuge tubes containing 1.4 mm tungsten beads and 1 mL of MEM with 20% FBS for one minute at 20 cycles per second using a Bead Ruptor 24 Elite (OMNI international, Kennesaw, GA, USA). The homogenates were clarified by centrifugation at 5000× *g* for 10 min at 4 °C, and the supernatant was used for RNA extraction and titration by plaque assay.

### 2.5. Viral Detection

Nucleic acid extraction from samples was performed using a commercial RNA Kit (Qiagen). For viral detection, a quantitative reverse transcriptase–polymerase chain reaction (RT-qPCR) assay protocol was established following primers and probes previously described for YFV by Domingo et al. (2012) [23]. The probe was modified to FAM and BHQ1. The following thermal conditions were used: 50 °C for 30 min; 95 °C for 2 min; 45 cycles of 95 °C for 15 s; 60 °C for 1 min. Only individuals who tested positive for the virus in the body were subsequently analyzed for YFV presence in their legs and saliva.

To detect infectious YFV virions, all samples were analyzed by plaque titration in 12-well plates of Vero C76 cells. Titration was performed as previously described by Medina et al. (2012) [24]. Briefly, 10-fold serial dilutions of each sample in MEM supplemented with 2% FBS and antibiotics were added to confluent Vero C76 monolayers and incubated for one hour with periodic gentle rocking to facilitate virus adsorption at 37 °C. The volume of the inoculant was 100 μL in each well. The plaques were incubated without disturbance for 4 days at 37 °C. Neutral red vital dye at 6% was used to visualize the plate.

### 2.6. Statistical Analysis

The mosquito specimens were examined to assess the infection rate (IR), dissemination rate (DR), and transmission rate (TR) of the virus. IR represents the proportion of mosquitoes with infected bodies among the total number of engorged mosquitoes. DR indicates the percentage of mosquitoes with infected legs among those with infected bodies. TR is recorded as the percentage of mosquitoes containing infectious virus in their saliva among those with disseminated infections.

The analysis was performed on 434 records of mosquitoes tested for body infection, belonging to two different colonies (La Plata and Tartagal) and at four different time points (3, 7, 14, and 21 DPI). The objective was to evaluate whether the infection percentage was affected by time using the mosquito colonies as variables. The same analysis was then applied to infection records in legs and saliva over those positive for body infection. Samples were considered infected if either RT-qPCR or titration was positive. In each case, generalized linear models for binomial response were applied [25], where the response *Y_i_* is the infection status of the individual in the part of interest (body, leg, or saliva in each case) and *p_i_* is the probability of infection. The model was then defined as follows: *logit*(*p*_*i*_) = *Colony* + *Time*, being *logit*(*p_i_*) = *ln*(*p_i_*/(1 − *p_i_*)). Due to the scarcity of samples in some combinations of colonies and time, interaction models are ruled out. The post-hoc analysis, in cases where the model retained a predictor, involved analyzing the confidence intervals of all possible pairs, with the confidence level adjusted using the Sidak method. Similar methodology was applied for analyzing viral titers. All statistical analyses were performed using R software v 4.3.3 [26]), and post-hoc analyses were conducted using the emmeans package [27].

## 3. Results

A total of 434 mosquitoes were fed with infected blood containing 7.9 log PFU/mL for the La Plata colony (n = 229), and 8.6 log PFU/mL for the Tartagal colony (n = 205). Successful infection was observed in both groups at each DPI tested, and the infection, dissemination, and transmission rates were calculated and analyzed (Table 1).

Taking into account the statistical analysis of body infection, the model predicted a significantly lower overall IR for Tartagal (14%) compared to La Plata (46%) (*p* value ˂ 0.001), while it did not record significant differences in pairwise comparisons between mosquito colonies among different DPIs. Viral dissemination was evident across all DPIs in both groups, showing a similar overall DR between them around 50% (Table 1). When leg infections were analyzed, the model did not identify the mosquito colony as a determining factor but rather highlighted the time variable. However, it did not record significant differences in the pairwise comparisons between DPIs. Despite this, the model predicted a probability of infection in the leg of 0.4 at 3 DPI and 0.7 at 21 DPI (Table 2), meaning it almost doubles, which would indicate an expected trend of increasing probability over time. However, TR could not be analyzed because the numbers were excessively low. The transmission rate was verified early at 3 DPI in La Plata despite the low number of individuals with viral dissemination, while, for Tartagal colony, it was not able to confirm at this time since only one female tested positive in legs, even though transmission for Tartagal was confirmed at 7 DPI. Considering 7, 14, and 21 DPI, the TR seems to increase with the time for the Tartagal colony, while it seems to remain more stable for the La Plata colony (Table 1).

The mean viral titers were calculated for each colony and DPI considering the titers that were possible to obtain (Table 3). The total means for La Plata and Tartagal are 4.4 and 4.7 log PFU/mL, respectively. The statistical analysis did not show significant differences between colonies, but the model predicted mean viral titers of the same order for 3 and 7 days post-infection (3.7 and 3.6 log PFU/mL, respectively) and a different second group for 14 and 21 days (5.0 and 5.4 log PFU/mL, respectively).

## 4. Discussion

Yellow fever is not an endemic disease in Argentina; even so, sylvatic YFV epizootic events have occurred in the country’s recent history. This fact, combined with the widespread presence of *Ae. aegypti* in the territory [28,29], highlights the risk of an urban outbreak when the necessary conditions arise. This risk is further increased by the fact that outbreaks in Brazil are progressively approaching our border over time.

There is knowledge about the vector competence of *Ae. aegypti* for transmitting viruses like dengue, zika, and chikungunya in Argentina [21,30], but studies related to the YFV are lacking. Therefore, we conducted tests to assess whether our local *Ae. aegypti* mosquitoes are competent vectors for transmitting this virus. Our results indicate that *Ae. aegypti* mosquitoes from both colonies are susceptible to YFV infection and transmission, but the probability of infection in the body is significantly higher for La Plata (0.46) than for Tartagal (0.14), while infection in the legs and saliva displayed similar rates between them. It seems that there is the presence of a midgut barrier; as, once this is overcome, the infection progresses without apparent differences. One possible explanation for the differences in infection rates between the two colonies is related to the infective process, which could be affected by virus–microbiota interactions [31], since microbiota vary greatly between local habitats [32]. Among these microorganisms, *Wolbachia* is well known for reducing viral infection in mosquitoes, but little evidence of natural infection in *Ae. aegypti* populations worldwide has been found [33,34]. Regarding the colonies evaluated in the present study, the Tartagal colony has been examined with negative results so far [33], as well as the La Plata colony, based on unpublished results (personal communication). In addition, previous studies for the dengue virus conducted on *Ae. aegypti* from Aguaray, near Tartagal (30 km), and La Plata also showed significant differences in vector competence with the overall infection rate being higher for La Plata than for Aguaray. However, for the chikungunya virus, significant differences were observed in the opposite direction: La Plata was refractory (5%) while Salta showed a moderate infection rate (37%). Moreover, in the same study, multilocus genotype analysis showed Aguaray as the most distinct among the studied *Ae. aegypti* populations. Genetic divergence is known to be a factor contributing to differences in vector competence among populations [30].

Bearing in mind the yellow fever vector competence in *Ae. aegypti* populations, several studies have been conducted throughout the world. Generally, laboratory procedures differ from natural conditions in aspects such as temperature fluctuations and feeding. These differences could influence the results regarding vector competence. Moroever, the protocols generally used are not standardized and have fluctuated over time in various aspects of the process, and the results obtained in different studies are difficult to compare [35]. Nevertheless, some observations can be made. General results showed a moderate infection rate of the YFV in *Ae. aegypti* with some exceptions in which high rates were observed between 7 and 14 DPI reaching 80% in mosquitoes from Australia and Brazil [17,36]. Complete susceptibility at 14 DPI has been observed only in a study in which the intrathoracic route of infection was used, as opposed to the oral infection generally used [20]. For Brazil, studies including individuals from different country states showed IR ranging 1–48% at 14 DPI [18,19], while a range of 0–21% at 20 DPI was observed in another study, which included *Ae. aegypti* populations from Africa and Central and North America, in which a triturated infected mouse brain was used as a viral inoculum, and viral detection was made using a fluorescent antibody assay [37]. Additionally, another study using *Ae. aegypti* from Africa and Central and South America showed IR at 14 DPI ranging 2–57%, with the highest rate corresponding to Africa [38].

Since most studies have been evaluated primarily at 14 DPI, there is little information regarding the infection rate at 3 DPI. Research on *Ae. aegypti* from Brazil found that the infection of 447 individuals was unsuccessful after that time [16]. Our results confirm that the Argentine *Ae. aegypti* population is not only capable of becoming infected but also of transmitting YFV at 3 DPI, at least with the strain locally isolated. Mosquitoes from both colonies evaluated showed infection at 3 DPI, higher in La Plata (53%) than in Tartagal (8%). The transmission rate was confirmed for La Plata at 3 DPI, while in Tartagal, the low number of mosquitoes with dissemination made it difficult to verify infection. Previous studies reported transmission rates only in some cases, as they primarily focused on infection rate levels. Couto and Lima (2017) and Van den Hurk (2011) reported transmission rates at 14 DPI, ranging from 0 to 96% among different populations tested, with the latter reporting a 0% transmission rate at 3 and 7 DPI [16,36].

Upon comparing viral detection methodologies, we observe a discrepancy between the number of samples that tested positive by real-time PCR and those that tested positive by plaque assay. This may be due to the fact that viral RNA detected by real-time PCR does not necessarily correspond to a fully infectious particle capable of forming a plaque [39]. In cases where it was possible to obtain a titer, it appears to be higher than the results reported by other authors. In a study where mosquitoes were infected with the blood of a similar titer (around 7), the authors observed a lower log PFU/mL at 14 DPI (around 3) compared to ours (around 5) [19], while similar titers were founded in similar studies but with zika virus [21].

Among the procedures used in vector competence studies to estimate transmission rates, salivation through capillaries is the most commonly used technique to avoid using laboratory animals as models. However, this method may underestimate the amount of virus present in the saliva and, consequently, the competence to transmit the virus [40]. Otherwise, it was seen that mosquitoes that came from a colony with a high number of laboratory generations could alter their susceptibility to viral infection [38]. This is the case with our colonies, where original mosquitoes were collected from different geographic areas of Argentina but had been colonized over several generations.

Despite our research probably underestimating the level of transmission, the results prove that the Argentinean *Ae. aegypti* population is able to transmit the YFV with an overall rate of 50%, including successful transmission at 3 DPI for the La Plata colony. Although the Tartagal colony had a lower infection rate, the transmission rate increased over the evaluated days, peaking at 21 DPI, while the La Plata colony maintained consistent numbers throughout the study, resulting in a similar overall transmission rate between colonies. Our work reveals important knowledge about what we might expect in a real urban outbreak in which the extrinsic incubation period is very short.

The reason why urban YFV outbreaks are not occurring nowadays is still unknown, but it is clear that several factors are present that increase the risk of occurrence in the region. First of all, the distribution of viral outbreaks in Brazil is getting closer to the Argentinean border, where urbanization is increasingly intruding on sylvan areas, and there are still unvaccinated people. Second, insufficient vector control strategies, along with short extrinsic incubation periods, are important risk factors that should not be underestimated.

This work is the first contribution to YFV vector competence studies in Argentina, involving mosquitoes originally from geographically distant cities and a locally isolated viral strain. It is now known that the Argentinean *Ae. aegypti* is competent in transmitting the YFV, but more studies are needed to understand other factors that could be affecting vector capacity.

## 5. Conclusions

Our main objective in this study was to evaluate the potential resurgence of urban YF outbreaks in the country through the study of the vector competence of *Ae. aegypti* from two colonies that originated from geographically distant cities in Argentina. Our results indicate that the Argentine population of *Ae. aegypti* is competent for YFV transmission. Moreover, mosquitoes originated in La Plata showed a significantly higher infection rate and an early transmission rate at 3 DPI. Although mosquitoes from Tartagal showed a lower infection rate, the transmission was proven at 7 DPI. These facts demonstrate that the Argentine *Ae. aegypti* population, under the test conditions, presents a short extrinsic incubation period. These findings highlight the need for an improved monitoring of vectors in urban environments. Additionally, more effective vector control programs are required to mitigate the risk of a resurgence of the YFV.

## Figures and Tables

**Figure 1 viruses-17-00718-f001:**
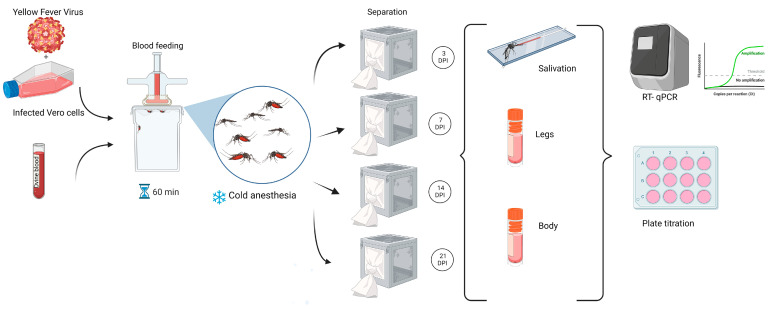
Workflow for experimental design: oral infection of adult *Aedes aegypti* mosquitoes, followed by detection procedures in different body parts using RT-qPCR and plaque titration.

**Table 1 viruses-17-00718-t001:** Tests for yellow fever virus infection were conducted on *Aedes aegypti* mosquitoes from two colonies originating in the cities of La Plata and Tartagal, Argentina. The tests were analyzed at 3, 7, 14, and 21 days post-infection (DPI). The numbers correspond to the mosquitoes that were fed and those that were infected in the body, legs, and saliva. Infection rates (IRs), dissemination rates (DRs), and transmission rates (TRs) are reported. The standard error is presented in parentheses.

Colony	DPI	Fed	Body	IR%	Legs	DR%	Saliva	TR%
La Plata	3	49	26	53	10	38	6	60
	7	53	25	47	12	48	4	33
	14	77	36	47	19	53	10	53
	21	50	19	38	15	79	8	53
	Total	229	106	46 (3.3)	56	53 (4.8)	28	50 (6.7)
Tartagal	3	25	2	8	1	50	0	0
	7	61	9	15	5	56	2	40
	14	57	5	9	4	80	3	75
	21	62	13	21	7	54	6	86
	Total	205	29	14 (2.4)	17	59 (9.1)	11	65 (11.6)

**Table 2 viruses-17-00718-t002:** Probability of infection in the body (IR) according to the colony and probability of infection in the legs (DR) predicted by the model. The 95% confidence interval (CI95) is shown.

**IR Model Predictions for Colonies**		
	Estimate	CI95
Tartagal	0.14	(0.095–0.205)
La Plata	0.46	(0.391–0.537)
**DR Model Predictions for DPIs**		
	Estimate	CI95
3 dpi	0.39	(0.198–0.629)
7 dpi	0.50	(0.299–0.702)
14 dpi	0.56	(0.369–0.737)
21 dpi	0.69	(0.46–0.851)

**Table 3 viruses-17-00718-t003:** Viral titers in bodies, legs, and saliva obtained after 3, 7, 14, and 21 days post yellow fever virus oral infection in *Aedes aegypti* individuals from La Plata and Tartagal, Argentina.

Colony	DPI	Mean Body Titer (log10 PFU/mL) ± SD (N)	Mean Leg Titer (log10 PFU/mL) ± SD (N)	Mean Saliva Titer (log10 PFU/mL) ± SD (N)
La Plata	3	3.77 ± 1.56 (19)	2.68 ± 1.03 (7)	2.70 (1)
	7	3.7 ± 0.97 (18)	2 ± 0.96 (8)	5.48 (1)
	14	4.84 ± 0.79 (27)	3.04 ± 0.67 (18)	1.48 ± 0.67 (2)
	21	5.25 ± 0.55 (13)	3.21 ± 0.43 (10)	1.80 ± 0.71 (2)
	Total	244		
Tartagal	3	3.31 ± 0.23 (2)	2.20 (1)	0
	7	3.34 ± 1.03 (8)	3.63 ± 0.31 (3)	0
	14	5.91 ± 0.53 (5)	3.60 ± 0.37 (4)	3.30 (1)
	21	5.56 ± 1.05 (7)	3.06 ± 0.75 (6)	1.70 (1)
	Total	205		

## Data Availability

Data are contained within the article.

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
