# Peer review of "Assessing Urban Yellow Fever Transmission Risk: Aedes aegypti Vector Competence in Argentina"

_viruses, 2025, doi:10.3390/v17050718_

Round 1

Reviewer 1 Report

Comments and Suggestions for Authors

Some of the environmental conditions of this study, while likely a practical necessity, may affect vector competence, so would be worth discussion.  Specifically:

Line 107 Rearing mosquitoes at constant temperatures when they would be subject to temperature fluctuations in nature

Line 107.  Providing Aedes aegypti with a constant sugar source (raisins) when in nature they are known to feed very little on sugar, preferring human blood.

Lines 110-112  Feeding Ae. aegypti ovine blood, when they vastly prefer human blood in nature

Line 224.   If yellow fever exists in a sylvatic cycle in Argentina, doesn't that make it endemic?

Lines 264-275.  If very few studies have demonstrated tranmission of YFV at 3dpi, that might be worth adding in the Abstract as a notable finding of this paper.

Line 293.  I think you mean "underestimating" instead of "sub-estimate"

Lines 301-307.  The question of why there is not more urban YF in Argentina is a very interesting one.  How many imported cases of YF are there from countries such as Brasil experiencing YFV epidemics?  A travelling viraemic case entering a city with an Aedes aegypti population would seem to be a very likely source of an outbreak.

Author Response

Response to Reviewer 1

Comments 1: Some of the environmental conditions of this study, while likely a practical necessity, may affect vector competence, so would be worth discussion.  Specifically:

  • Line 107 Rearing mosquitoes at constant temperatures when they would be subject to temperature fluctuations in nature
  • Line 107.  Providing Aedes aegypti with a constant sugar source (raisins) when in nature they are known to feed very little on sugar, preferring human blood.
  • Lines 110-112 Feeding aegypti ovine blood, when they vastly prefer human blood in nature

Line 269: the following sentence were added:

“Generally, laboratory procedures differ from natural conditions in aspects such as temperature fluctuations and feeding. These differences could influence the results regarding vector competence. “

Comments 2: Line 224.   If yellow fever exists in a sylvatic cycle in Argentina, doesn't that make it endemic?

In Argentina, the sylvatic cycle is maintained between mosquitoes and non-human primates. It was described in 1966 and the last description was made in 2007-2009. The circulation of the yellow fever virus in Argentina is sporadic and depends on the yellow wave that comes from Brazil. At least at present it is described that way.

There is no continuous circulation between non-human primates and mosquitoes, and cases in the population are imported, except in outbreak situations.
Since June 2017, and based on the evidence presented by the country, the Scientific-Technical Advisory Group on Geographic Mapping of Yellow Fever Risk (GRYF) considers that Argentina is outside the endemic zone of yellow fever.

Comments 3: Lines 264-275.  If very few studies have demonstrated transmission of YFV at 3dpi, that might be worth adding in the Abstract as a notable finding of this paper.

Line 28: A sentence was added: “Regarding the infection timeline, we observed a very early infection in the La Plata colony at 3 DPI, in contrast to previous studies.”

Comments 4: Line 293.  I think you mean "underestimating" instead of "sub-estimate".

Line 315: The word sub-estimate was replaced by underestimating

Comments 5: Lines 301-307.  The question of why there is not more urban YF in Argentina is a very interesting one.  How many imported cases of YF are there from countries such as Brasil experiencing YFV epidemics?  A travelling viraemic case entering a city with an Aedes aegypti population would seem to be a very likely source of an outbreak

The last reported cases were in 2018, when seven cases were recorded with a history of travel to Brazil and no vaccination history according to the national epidemiology bulletin. However, we believe that they may not necessarily have been the only ones. This number could be higher if a traveler was asymptomatic or if they had symptoms but did not go to a medical service and, ultimately, if the medical service diagnosed them incorrectly.

Reviewer 2 Report

Comments and Suggestions for Authors

The method of oral infection of the mosquitoes appears to be sound. However, the description on line 105 refers to Bonica et al. 2019 but fails to mention that this is reference 21.

The RT-qPCR method used to measure the presence of viral RNA is based on Domingo et al. 2012. The original method used a probe with FAM and Tamra. Was this configuration used or was the Tamra replaced by BHQ-1 and were the cycling parameters identical. It may be best if these details were added to the paper.

It is reasonable that the interaction with the microbiota of the mosquitoes should be mentioned and reference 31 summarises this quite well. As Wolbachia is being increasingly used to reduce the transmission of viruses such as yellow fever it may have been interesting to check the two colonies for the presence of this organism using a PCR such as that used by (Zhang et al., 2022).

It appears that Wolbachia infected Aedes aegypti are being introduced in Brazil in regions close to Argentina.

The call for improved vector surveillance as outlined in line 323 is admirable. However, it may be time to also start urging the implementation of control programs to help prevent further outbreaks.

ZHANG, H., GAO, J., MA, Z., LIU, Y., WANG, G., LIU, Q., DU, Y., XING, D., LI, C., ZHAO, T., JIANG, Y., DONG, Y., GUO, X. & ZHAO, T. 2022. Wolbachia infection in field-collected Aedes aegypti in Yunnan Province, southwestern China. Front Cell Infect Microbiol, 12, 1082809.

Author Response

Comments 1: The method of oral infection of the mosquitoes appears to be sound. However, the description on line 105 refers to Bonica et al. 2019 but fails to mention that this is reference 21.

Line 107: Quote [21] was added to the manuscript

Comments 2: The RT-qPCR method used to measure the presence of viral RNA is based on Domingo et al. 2012. The original method used a probe with FAM and Tamra. Was this configuration used or was the Tamra replaced by BHQ-1 and were the cycling parameters identical. It may be best if these details were added to the paper.

The suggestion was added from line 151 in the following sentence: "The probe was modified to FAM and BHQ1. Under the following thermal conditions: 50 °C for 30 min; 95 °C for 2 min 45 cycles of 95 °C for 15 s, 60 °C for 1 min"

Comments 3: It is reasonable that the interaction with the microbiota of the mosquitoes should be mentioned and reference 31 summarizes this quite well. As Wolbachia is being increasingly used to reduce the transmission of viruses such as yellow fever it may have been interesting to check the two colonies for the presence of this organism using a PCR such as that used by (Zhang et al., 2022).

Line 307: The following sentence was added:  “Among these microorganisms, Wolbachia is well known for reducing viral infection in mosquitoes, but little evidence of natural infection in Ae. aegypti populations worldwide has been found [33, 34]. Regarding the colonies evaluated in the present study, the Tartagal colony has been examined with negative results so far [33], as well as the La Plata colony, based on unpublished results (personal communication).”

Comment 4: It appears that Wolbachia infected Aedes aegypti are being introduced in Brazil in regions close to Argentina.

Your comment is very accurate. It would be interesting to see if in the future these mosquitoes with Wolbachia have any impact on the Argentine populations for YFV as well as other arboviruses.

Comments 5: The call for improved vector surveillance as outlined in line 323 is admirable. However, it may be time to also start urging the implementation of control programs to help prevent further outbreaks.

Line 345: The mentioned line was rewritten according to the suggestion as follows “These findings highlight the need for improved monitoring of vectors in urban environments. Additionally, more effective vector control programs are required to mitigate the risk of a resurgence of YFV”
